# Mito-SiPE is a sequence-independent and PCR-free mtDNA enrichment method for accurate ultra-deep mitochondrial sequencing

Darren J. Walsh [1,2], David J. Bernard [1], Faith Pangilinan[1], Madison Esposito[1], Denise Harold[2], Anne Parle-McDermott[2] & Lawrence C. Brody [1✉]

The analysis of somatic variation in the mitochondrial genome requires deep sequencing of mitochondrial DNA. This is ordinarily achieved by selective enrichment methods, such as PCR amplification or probe hybridization. These methods can introduce bias and are prone to contamination by nuclear-mitochondrial sequences (NUMTs), elements that can introduce artefacts into heteroplasmy analysis. We isolated intact mitochondria using differential centrifugation and alkaline lysis and subjected purified mitochondrial DNA to a sequence-independent and PCR-free method to obtain ultra-deep (>80,000X) sequencing coverage of the mitochondrial genome. This methodology avoids false-heteroplasmy calls that occur when long-range PCR amplification is used for mitochondrial DNA enrichment. Previously published methods employing mitochondrial DNA purification did not measure mitochondrial DNA enrichment or utilise high coverage short-read sequencing. Here, we describe a protocol that yields mitochondrial DNA and have quantified the increased level of mitochondrial DNA post-enrichment in 7 different mouse tissues. This method will enable researchers to identify changes in low frequency heteroplasmy without introducing PCR biases or NUMT contamination that are incorrectly identified as heteroplasmy when long-range PCR is used.

[1] Gene and Environment Interaction Section, National Human Genome Research Institute, NIH, Bethesda, MD, USA. [2] School of Biotechnology, Dublin City University, Dublin, Ireland. ✉email: lbrody@nih.gov

Decades of research have established a link between mitochondrial DNA variation and human health. Recent advances in DNA sequencing technologies have led to an increased ability to interrogate the mitochondrial genome for low-frequency mutations associated with various disease states. Mitochondrial DNA mutations have been associated with ageing[1] and a myriad of disease phenotypes[2]. Disorders caused by inherited and acquired mitochondrial DNA variants affect ~1 in 4300 of the population[3]. These variants were initially thought to solely originate from the matrilineal inheritance of mitochondrial DNA molecules, however more recent studies have shown that somatic mutations also occur in mtDNA over time in a tissue-specific manner[4–6]. There are hundreds to thousands of mitochondrial DNA molecules in every human cell. This number is dependent on the tissue, cell type and energy state of the mitochondria[7]. The fluctuating, multi-copy nature of mitochondrial DNA means that mutations can be present at any frequency within a cell, unlike the diploid nuclear genome. The presence and frequency of mitochondrial DNA mutations is referred to as mitochondrial heteroplasmy.

Mitochondrial heteroplasmy has been increasingly investigated as a contributor to human disease. To date, it has been linked to various diseases, including cardiomyopathy, hypertension, epilepsy, Parkinson's disease and optic neuropathy[8–12]. Elevated heteroplasmy levels have also been linked to tumour aggressiveness and poor cancer prognoses[13,14]. These studies demonstrate that mitochondrial heteroplasmy analysis could help to identify unknown molecular mechanisms that drive some disease states. Additionally, evidence suggests that mitochondrial DNA heteroplasmy could be used as a potential target for diagnosis/prognosis of particular conditions and even perhaps as a therapeutic target[15]. Given these findings, it is important that mitochondrial heteroplasmy, particularly low-frequency heteroplasmy, can be identified and quantified as a part of disease-related studies. Such investigations require high to ultra-high sequencing coverage (>1000X–>10,000X) of the mitochondrial genome to reliably quantitate low-frequency heteroplasmy with a high degree of sensitivity and specificity. Currently, this is typically achieved by using probe hybridisation or long-range polymerase chain reaction (PCR) to enrich mitochondrial DNA.

Probe Hybridisation[16] uses complementary probes that bind mitochondrial sequences to separate mtDNA from nuclear DNA. Another approach is long-range PCR[17] which amplifies the mitochondrial genome, typically in two, overlapping fragments. Both probe hybridisation and long-range PCR amplification require complementary binding of probes/primers to enrich mtDNA from whole DNA extracts. The widespread use of these methodologies in many heteroplasmy studies is due to their ease, amenability to high-throughput processes and efficacy in producing mtDNA appropriate for ultra-deep sequencing[4,5,18,19]. These sequence-dependent methods are imperfect. Probes and primers designed to match reference alleles may select against rare heteroplasmic variants that are of interest. Additionally, PCR amplification is known to introduce errors that may appear as false positive heteroplasmic variants. Arguably, the most problematic issue for these techniques is the contamination of nuclear-mitochondrial elements (NUMTs)[20].

NUMTs are nuclear sequences that share high levels of sequence identity with mtDNA. They had arisen from the somatic translocation of mitochondrial DNA into the nuclear genome. The number, size and sequence of NUMTs varies within species[21], including between human populations and individuals[22]. The entire mitochondrial genome is represented in the human nuclear genome[22]. As a result of this, it is extremely difficult to design primers or hybridisation probes that will selectively enrich mitochondrial DNA without also enriching NUMT sequences. Multiple studies have found that NUMT contamination was present in mtDNA sequencing data that used either probe hybridisation or PCR amplification to enrich mtDNA[23–26]. Most notably, NUMT contamination is thought to explain the apparent paternal inheritance of mitochondrial DNA in humans that was reported in PNAS in 2018[27–30]. The difficulty posed by deciphering NUMT contamination from true mitochondrial DNA may require the use a sequence-independent methods for mitochondrial enrichment.

Here, we have adapted a previously described[31] sequence-independent and PCR-free technique which relies on differential centrifugation and alkaline lysis to separate mitochondria from other tissue/cellular debris. We have called this method Mito-SiPE (a sequence-independent, PCR-free mitochondrial DNA enrichment). We provide evidence that this method can be effectively used to isolate mitochondrial DNA from different tissues for subsequent mtDNA sequencing, achieving ultra-deep coverage of the mitochondrial genome when combined with an appropriate NGS data pipeline.

A sequence-independent method for mitochondrial enrichment carried out on blood and cell culture samples was previously described using an exonuclease digest[32]. Its use on samples with a modest amount of starting material offers a unique approach for measuring heteroplasmy in scarce samples such as cultured cells. However, compared to Mito-SiPE, the method yielded a reduced mapping efficiency and sequencing depth on average, limiting its use in low-frequency heteroplasmy analysis.

*Polg* mutator mice are used as positive controls to compare Mito-SiPE with long-range PCR amplification enrichment of mtDNA. These mice lack the ability to 'proof-read' their mitochondrial DNA and, as a result, gather single nucleotide mutations at a higher frequency than their wild-type counterparts[33]. We propose that this method can be applied to a range of species to allow researchers to reliably investigate mitochondrial heteroplasmy and expand our current knowledge of the contribution of somatic mitochondrial mutations to human ageing and disease. This methodology negates the impact of NUMT contamination and PCR error introduction when assessing heteroplasmy and is, therefore, more sensitive and accurate than long-range PCR amplification.

## Results

**Mito-SiPE produces highly enriched mitochondrial DNA.** This technique for enriching mtDNA for sequencing is rooted in classic cell biology and biochemistry methods for subcellular fractionation. It utilises a combination of differential centrifugation and alkaline lysis to separate the mitochondria from nuclear and cytoplasmic cellular components. The preparation is then used to purify mitochondrial DNA with minimal contamination from nuclear DNA (Fig. 1a). To test this method, we used it on seven different mouse tissues; brain, heart, lung, kidney, liver, spleen and muscle. Mitochondrial copy number was assessed *via* quantitative polymerase chain reaction (qPCR) in enriched samples from the seven tissues and compared to unenriched, whole DNA extracts (Fig. 1b). qPCR was used to calculate the ratio of mitochondrial DNA to nuclear DNA, which provides an estimate of the average mtDNA copy number per cell. A significant increase (100–1000-fold, $P < 0.0005$) in mitochondrial DNA copy number was observed in samples that underwent enrichment. This effect was present across the seven tissues of interest (Fig. 1c).

The performance of this technique to produce pure, high-quality mtDNA for next-generation DNA sequencing was assessed by applying the method to 163 samples across seven different mouse tissues. These samples were then sequenced

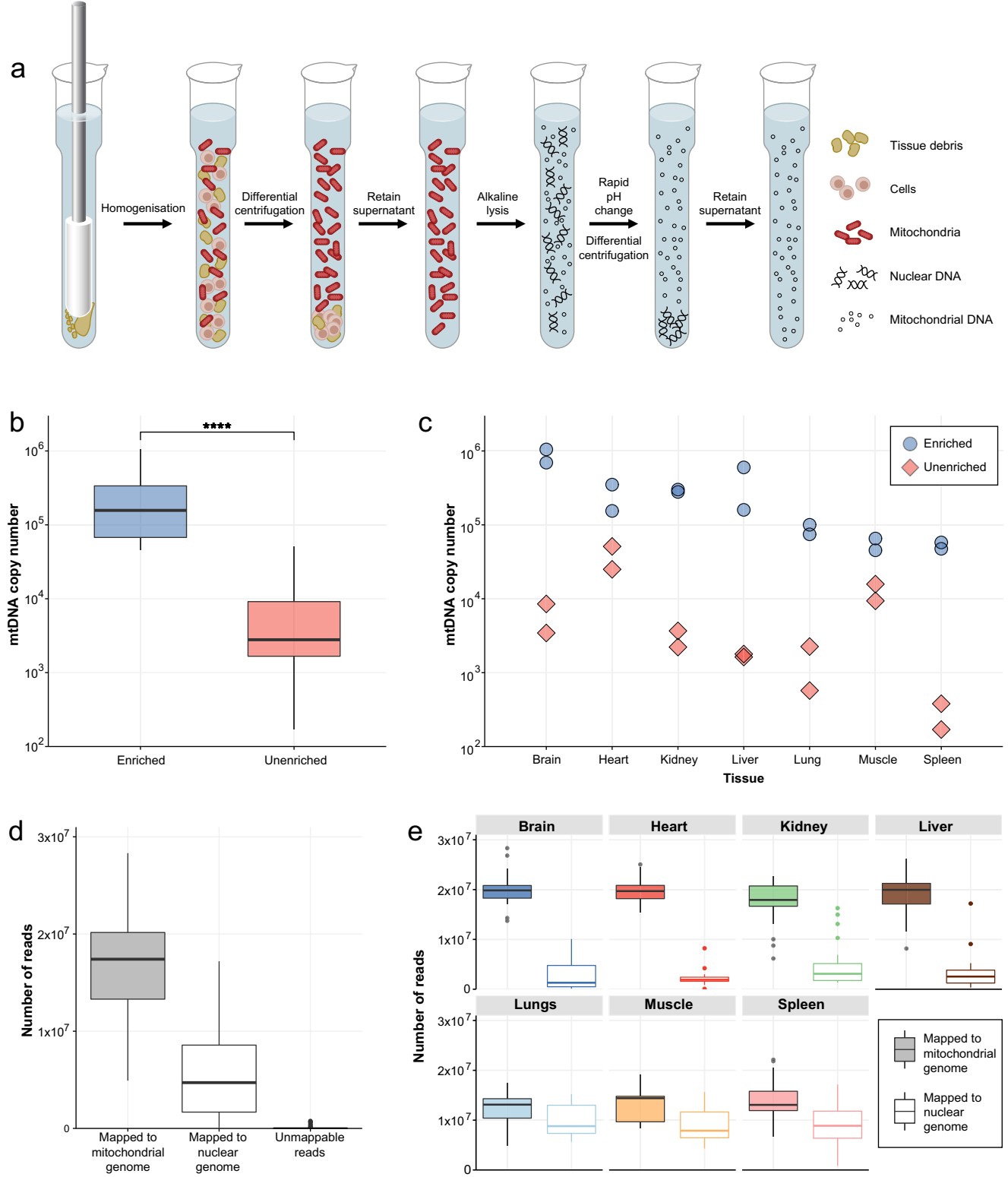

across four lanes of the Illumina NovaSeq platform. After quality control, alignment and removal of duplicates, the number of mitochondrial, nuclear and unmapped reads were assessed for each sample (Fig. 1d and Table 1). Of the mapped reads, an average of 75 ± 20% were mapped to the mitochondrial genome and 25 ± 20% to the nuclear genome. Unmappable reads made up 0.26 ± 0.63% (Table 2). The level of nuclear contamination in samples originating from the brain, heart, kidney and liver was

extremely low, with higher levels found in the lung, muscle and spleen (Fig. 1e).

**Mito-SiPE requires an alternative alignment strategy.** Two alignment strategies were used to assess the coverage of the mitochondrial genome and the distribution of nuclear contamination (Fig. 2a). The first method aligned filtered reads to the

**Fig. 1 PCR-free enrichment of mitochondrial DNA using differential centrifugation and alkaline lysis. a** Overview of sequence-independent and PCR-free mitochondrial DNA enrichment workflow. Homogenisation and differential centrifugation are used to enrich mitochondria. Alkaline lysis is then used to isolate mitochondrial DNA from any remaining nuclear DNA. **b** Relative quantification of mtDNA copy number from seven different mouse tissues that underwent enrichment. The mitochondrial DNA to nuclear DNA ratio (mtDNA:nuDNA) was assessed via qPCR. $P < 0.0005$ Wilcoxon signed-rank test. Log scaled. The upper and lower hinges of the boxplot represent the 75th and 25th percentiles, respectively. The middle hinge represents the median. The upper and lower whiskers extend no further than 1.5x the adjacent interquartile range. **c** The mtDNA:nuDNA ratio across seven different tissues which underwent enrichment. ($n = 2$) for each tissue (brain, heart, lungs, liver, kidney, spleen and muscle). **d** The distribution of nuclear, mitochondrial and unmappable reads generated in each sample is displayed as boxplots ($n = 163$). The upper and lower hinges of the boxplot represent the 75th and 25th percentiles, respectively. The middle hinge represents the median. The upper and lower whiskers extend no further than 1.5x the adjacent interquartile range. **e** The distribution of total and mapped reads across seven mouse tissues after mtDNA enrichment; Brain ($n = 26$), Heart ($n = 26$), Kidney ($n = 25$), Liver ($n = 21$), Lungs ($n = 26$), Muscle ($n = 12$) and Spleen ($n = 27$). The upper and lower hinges of the boxplot represent the 75th and 25th percentiles, respectively. The middle hinge represents the median. The upper and lower whiskers extend no further than 1.5x the adjacent interquartile range.

**Table 1 The number of mitochondrial, nuclear and unmapped reads per tissue.**

| Tissue | Number of reads (millions) | | |
| --- | --- | --- | --- |
| | Mitochondrial (sd) | Nuclear (sd) | Unmapped (sd) |
| Brain | 20.027 (3.285) | 2.575 (2.729) | 0.073 (0.178) |
| Heart | 19.636 (2.166) | 2.188 (1.481) | 0.042 (0.099) |
| Kidney | 17.428 (4.224) | 4.783 (4.344) | 0.012 (0.021) |
| Liver | 18.931 (4.071) | 3.291 (3.774) | 0.081 (0.171) |
| Lungs | 12.206 (2.938) | 9.860 (3.137) | 0.057 (0.134) |
| Muscle | 13.146 (3.467) | 9.093 (3.589) | 0.156 (0.252) |
| Spleen | 13.813 (3.827) | 9.301 (4.368) | 0.037 (0.072) |
| All tissues | 16.641 (4.586) | 5.700 (4.660) | 0.057 (0.139) |

**Table 2 The percentage of reads aligned to the mitochondrial genome, nuclear genome and unmappable reads.**

| Tissue | Percentage of total reads | | |
| --- | --- | --- | --- |
| | Mitochondrial (sd) | Nuclear (sd) | Unmapped (sd) |
| Brain | 88.64 (11.09) | 11.02 (11.20) | 0.34 (0.85) |
| Heart | 89.93 (6.19) | 9.87 (6.23) | 0.21 (0.50) |
| Kidney | 78.68 (18.60) | 21.26 (18.63) | 0.06 (0.10) |
| Liver | 85.14 (15.36) | 14.52 (15.43) | 0.33 (0.63) |
| Lungs | 55.30 (13.21) | 44.44 (13.30) | 0.26 (0.60) |
| Muscle | 58.90 (15.02) | 40.39 (14.98) | 0.71 (1.17) |
| Spleen | 60.15 (16.53) | 39.69 (16.54) | 0.16 (0.32) |
| All tissues | 74.64 (19.76) | 25.10 (19.78) | 0.26 (0.63) |

whole genome and then isolated mitochondrial-aligned reads. The second method aligned all the reads to the mitochondrial genome first, then mapped any remaining unaligned reads to the nuclear genome. Coverage across the mitochondrial genome and distribution of nuclear contamination were assessed after duplicate removal. Average coverage across the mitochondrial genome exceeded 50,000X at each base pair and was dependent on the tissue of origin (Fig. 2b). Lung, muscle and spleen had lower levels of coverage due to the higher levels of reads mapped to nuclear DNA in these samples. Mapping to the whole genome caused a loss in coverage between nucleotide positions 7500–11,000 (Fig. 2b, top). This dip in coverage was not observed when reads were first aligned to the mitochondrial genome using the second alignment strategy (Fig. 2b, bottom). The latter method did not lead to an overall increase in sequencing coverage across the rest of the mitochondrial genome, indicating that nuclear reads were not misaligned to the mitochondrial genome when this method was used.

As this mtDNA enrichment method is sequence-independent, any reads that originate from nuclear DNA should have been randomly distributed throughout the whole genome. To test this hypothesis, the distribution of nuclear contamination was assessed using both alignment strategies (Fig. 2c). When reads were aligned to the whole genome, nuclear contamination appeared to be evenly distributed across the nuclear genome as expected, except for chromosome 1 (Fig. 2c, top). Aligning the reads to the mitochondrial genome first did not produce this same anomaly (Fig. 2c, bottom and Supplementary Fig. 1). This is due to a NUMT found on chromosome 1 of the reference genome that shares 99.94% sequence identity of its homologous sequence in the mitochondrial genome (Supplementary Fig. 2). As a result, the alignment tool (bwa) mapped ~50% of reads originating from this region to the mitochondrial genome and ~50% incorrectly, to the homologous NUMT on chromosome 1 when reads were mapped to the whole genome. This artefact was eliminated when reads were mapped to the mitochondrial genome first. Chromosomes 2 and 9 also had slightly elevated levels of mapped reads when compared to the rest of the genome. This effect was produced by both alignment strategies. However, further investigation showed that this was caused by the alignment of highly repetitive reads from across the genome to the same loci on chromosomes 2 and 9 (Supplementary Figs. 3–6). Mapping to the mitochondrial genome first appeared to be more effective at mapping true mitochondrial reads correctly, without increasing levels of spurious alignments. This alignment methodology was used to assess heteroplasmy levels in the tissues of $Polg^{D257A/D257A}$ and $Polg^{wt/wt}$ mice.

**Heteroplasmy analysis comparison between Mito-SiPE and lrPCR in *Polg* mutator mice.** There was an average of $7.15 \times 10^7$ reads produced per sample across all studied groups (Table 3; $n = 48$). A higher average number of reads was observed in the lrPCR amplification samples ($7.22$ and $7.19 \times 10^7$, $Polg^{D257A/D257A}$ ($n = 12$) and $Polg^{wt/wt}$ ($n = 12$), respectively) compared to the mtDNA prep samples ($7.15$ and $7.02 \times 10^7$ $Polg^{D257A/D257A}$ and $Polg^{wt/wt}$, respectively). The proportion of reads mapped to the mitochondrial genome was also higher in the lrPCR amplification samples than in the mtDNA preparation (Mito-SiPE) samples (Table 3). The average coverage across the mitochondrial genome was higher in the long-range PCR amplification samples (Average depth 137,000X compared to mtDNA prep average of 123,000X). However, these samples had a loss in coverage towards the end of the two overlapping fragments (Fig. 3a, left). The mitochondrial DNA prep samples, broadly, had uniform coverage across the whole mitochondrial genome compared to the lrPCR samples. $Polg^{D257A/D257A}$ samples displayed minor region-specific fluctuations in coverage (Fig. 3a, right). This effect was not observed in the long-range PCR amplification assay (Fig. 3a, middle).

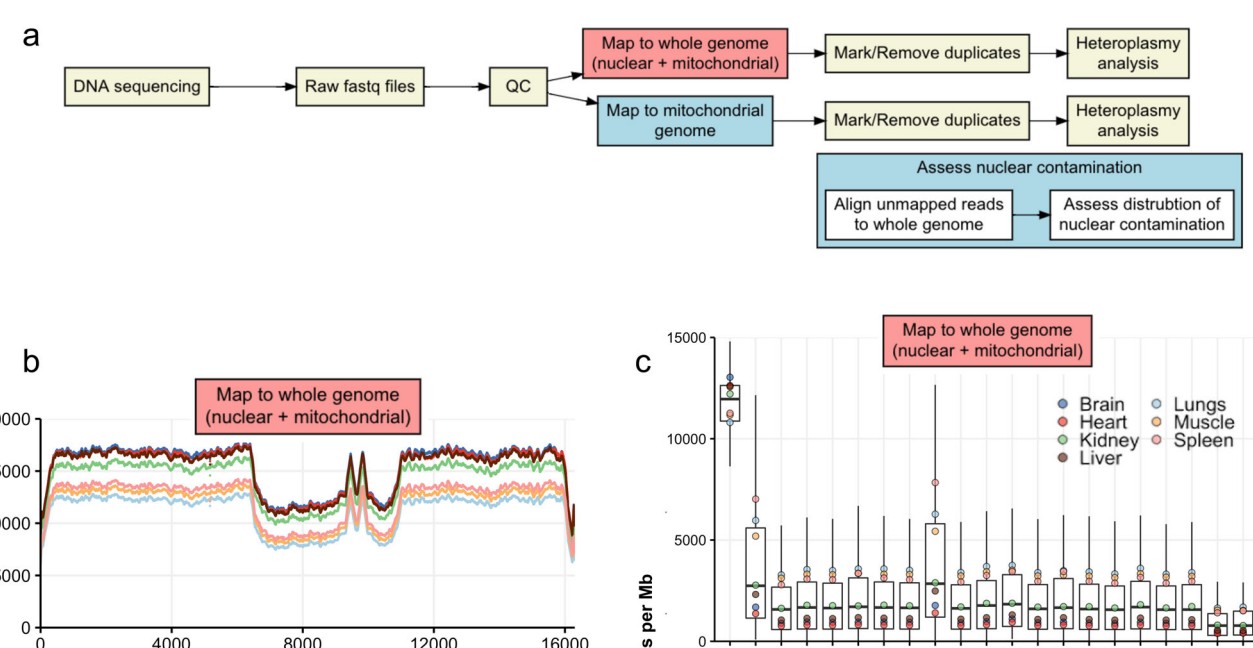

**Fig. 2 Alignment of sequencing data to whole-genome reference leads to misalignment of mitochondrial reads.** Contaminating nuclear reads are randomly distributed across the whole genome. **a** Overview of two methods for mapping sequencing data; *via* whole genome mapping or mapping exclusively to the mitochondrial genome followed by mapping of unaligned reads to the nuclear genome. **b** The sequencing coverage across the mitochondrial genome of seven mouse tissues when reads are mapped to the whole reference genome compared to when reads are mapped exclusively to the mitochondrial reference genome. A 'hole' in coverage is observed between nucleotide positions 7500–11,000 when reads are aligned to the whole reference genome. **c** The boxplots represent the distribution of reads mapped per megabase of DNA to each chromosome when reads are aligned to the whole reference genome in contrast to when reads are aligned exclusively to the mitochondrial genome. The points represent the average reads per Mb mapped for each tissue. The upper and lower hinges of the boxplot represent the 75th and 25th percentiles, respectively. The middle hinge represents the median. The upper and lower whiskers extend no further than 1.5x the adjacent interquartile range.

**Table 3 The average total number of reads, mapped reads and sequencing depth for each methodology and genotype.**

| Assay type | Genotype | Total reads | | Mapped reads | | Coverage | |
|---|---|---|---|---|---|---|---|
| | | Average ($10^7$) | SD | Average ($10^7$) | SD | Average ($10^5$) | SD |
| lrPCR amplification | $Polg^{D257A}$ | 7.22 | 0.561 | 7.19 (99.5%) | 0.50 | 1.39 | 0.417 |
| mtDNA prep | $Polg^{D257A}$ | 7.15 | 0.848 | 3.98 (55.7%) | 1.86 | 1.22 | 0.245 |
| lrPCR amplification | $Polg^{wt}$ | 7.19 | 0.687 | 7.08 (98.5%) | 0.693 | 1.36 | 0.411 |
| mtDNA prep | $Polg^{wt}$ | 7.02 | 0.687 | 4.53 (64.5%) | 1.79 | 1.25 | 0.126 |

Three metrics were measured to assess heteroplasmy levels in $Polg^{D257A/D257A}$ and $Polg^{wt/wt}$ tissues: the number of heteroplasmic sites, the average alternative allele frequency (average heteroplasmy) and cumulative heteroplasmic burden. These three metrics have previously been reported in the literature[4,14,34–37]. The number of heteroplasmic sites is the number of nucleotide positions at which an alternative allele was identified above the threshold frequency (0.2%) in a sample. Alternative allelic calls

caused by sequencing error are present below this frequency and are indistinguishable from low-frequency heteroplasmy. Average heteroplasmy is the mean frequency of all variants observed in a sample above the threshold frequency. Finally, the cumulative heteroplasmic burden is the sum of all variant frequencies that were identified above the threshold frequency in a sample.

More heteroplasmic sites were observed in the $Polg^{D257A/D257A}$ mice than wild-type, in the brain, liver and kidney (Fig. 3b, top).

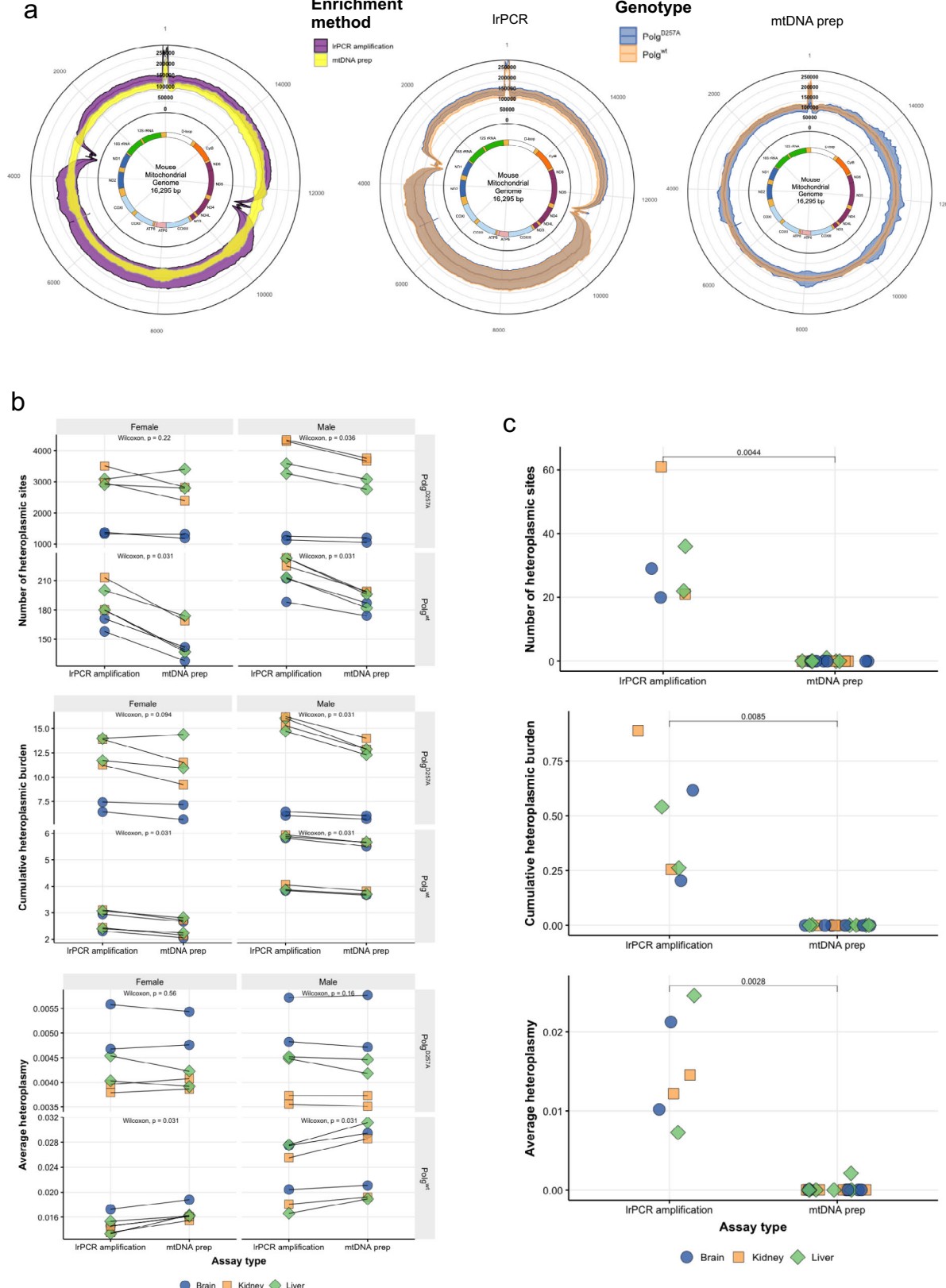

No difference was observed between $Polg^{D257A/D257A}$ males and females, however, there was a significant difference between sexes of $Polg^{wt/wt}$ (Fig. 3b, top, left panels vs right panels). There were significantly fewer heteroplasmic sites found in mtDNA preps of both male and female $Polg^{wt/wt}$ as well as $Polg^{D257A/D257A}$ males compared to lrPCR enrichment. This effect was not statistically

significant in female $Polg^{D257A/D257A}$ tissues. The difference between enrichment assays was larger in $Polg^{D257A/D257A}$ samples than in $Polg^{wt/wt}$ (Supplementary Fig. 7). When examining cumulative heteroplasmic burden, similar overall results were observed (Fig. 3b, middle). A lower burden was detected in mtDNA prep enriched samples compared to lrPCR in the same

**Fig. 3 Mitochondrial DNA preparations outperform long-range PCR amplification and reduce the impact of PCR errors and NUMT contamination on mitochondrial heteroplasmy. a** Read depth across the mitochondrial genome after sequencing and alignment for each enrichment method. The average sequencing depth was higher in the lrPCR (purple) samples than in those enriched using the mtDNA preparation method (yellow). There was a significant reduction of coverage towards the end of each amplicon fragment in the lrPCR samples. This reduction was not observed in the mtDNA prep samples. There were differences in the average sequencing depth between $Polg^{wt/wt}$ (orange) and $Polg^{D257A/D257A}$ (blue) using mtDNA prep (left) and lrPCR (right) methodologies. The sequencing depth of $Polg^{D257A/D257A}$ tissues enriched using the mtDNA prep method showed a region-specific variation in coverage, however, the average coverage remained comparable between both genotypes. This region-specific pattern was not observed in samples enriched using lrPCR. The standard deviations were larger in lrPCR-enriched samples, with one fragment showing larger differences than the other. This was likely due to PCR efficiency or variations due to attempts to mix both fragments in equimolar ratios. **b** The number of heteroplasmic sites (alternative allele frequency ≥0.2%), cumulative heteroplasmic burden and average heteroplasmy quantified in each sample. $Polg^{D257A}$ tissues had more heteroplasmic sites than $Polg^{wt}$. There were significantly less heteroplasmic sites observed in mtDNA prep samples of $Polg^{wt}$ males and females, and in $Polg^{D257A}$ males than in lrPCR-enriched samples. This effect was not significant in the $Polg^{D257A}$ females. Cumulative heteroplasmic burden displayed a similar pattern to the number of heteroplasmic sites; significantly lower levels were detected in $Polg^{wt}$ males and females and in $Polg^{D257A}$ males. The difference was not significant in $Polg^{D257A}$ females. Average alternative allele frequency displayed a different pattern of results compared to the previous two heteroplasmy metrics. $Polg^{D257A}$ tissues had lower mean alternative allele frequencies than $Polg^{wt}$. Significantly higher mean alternative allele frequencies were observed in lrPCR tissues from both male and female $Polg^{D257A}$ mice. Conversely, $Polg^{wt}$ samples enriched using the mtDNA prep methodology had higher mean alternative allele frequency than those enriched via lrPCR. Tissues are highlighted by colour; each line represents the same tissue that was enriched using both methodologies. Statistical comparisons between lrPCR and mtDNA prep were performed using a Wilcoxon signed-rank test. There were 24 lrPCR samples and 24 mtDNA prep samples ($Polg^{wt}$ $n = 24$, $Polg^{D257A}$ $n = 24$). Two samples for each tissue, sex and genotype were analysed. **c** Heteroplasmy levels of C57BL6 wild-type mice using both lrPCR and mtDNA prep enrichment methods. Mitochondrial DNA from C57BL6 tissues that were enriched using lrPCR had substantially higher levels of heteroplasmy across the three assessed metrics: number of heteroplasmic sites, cumulative heteroplasmic burden and average heteroplasmy. Unlike the Polg mutator mouse comparisons, these enrichments were not performed on the same tissues however the mice were the same sex and age at sacrifice. Statistical comparisons between lrPCR and mtDNA prep were performed using a Student's $t$-test. There were six samples in the lrPCR group and twelve samples in the mtDNA prep group. The mitochondrial genomes in the centre of the graph were created with BioRender.com.

three of four comparisons. Average heteroplasmy, however, displayed contrasting results (Fig. 3b, bottom). $Polg^{D257A/D257A}$ mice had lower average heteroplasmy levels than $Polg^{wt/wt}$ mice. In $Polg^{D257A/D257A}$ mice, the mean alternative allele frequency was significantly lower in mtDNA preps than in lrPCR enrichment. $Polg^{wt/wt}$ mice appeared to have higher average heteroplasmy levels, which were elevated in mtDNA preps compared to lrPCR amplification enrichment.

$Polg^{wt/wt}$ mice have a baseline of mitochondrial DNA mutation much higher than that of true wild-type C57BL6 mice (169 ± 29.4 and 0.06 ± 0.24 heteroplasmic sites, respectively). This is because heterozygous female breeders have an intermediate phenotype. Long-range PCR amplification enrichment was performed on total DNA extracted from the brain, kidney and liver of two wild-type C57BL6/J males. These enrichments were compared to the mtDNA preparation of six control C57BL6 males. These mice were age-matched. There was a significant and substantial increase in the number of heteroplasmic sites, cumulative heteroplasmic burden and average heteroplasmy in the lrPCR samples compared to samples that underwent mtDNA preparation (Fig. 3c).

Mitochondrial DNA prep enrichment was performed on seven tissues of $Polg^{wt/wt}$ and $Polg^{D257A/D257A}$ mice. Mutant mice displayed significantly higher levels of mitochondrial DNA mutation than wild-type across all tissues (Supplementary Fig. 8). Kidney, liver, colon, heart and lung had a similar number of heteroplasmic sites in $Polg^{D257A/D257A}$ mice with higher levels observed in spleen and lower levels found in the brain (Supplementary Fig. 9a). Colon and spleen had higher levels of average heteroplasmy than the other five tissues in $Polg^{D257A/D257A}$ mice (Supplementary Fig. 9b). Cumulative heteroplasmic burden was higher in $Polg^{D257A/D257A}$ colon and spleen tissues and lower in the brain than in heart, kidney, liver and lung (Supplementary Fig. 9c). There was no significant difference between the tissues of $Polg^{wt/wt}$ mice across any of the three heteroplasmy metrics that were assessed.

Analysis of the mutation profile observed using both lrPCR and Mito-SiPE showed similar results in $Polg^{wt/wt}$ and

$Polg^{D257A/D257A}$ (Supplementary Fig 10a). $Polg^{D257A/D257A}$ mice had higher levels of mutations occurring at 'C' nucleotide positions in the reference genome (light-strand). Interestingly, $Polg^{wt/wt}$ had more mutations at the 'A' nucleotide position, indicating that perhaps there is some selection that occurs when mutations are passed from $Polg^{D257A/wt}$ to their progeny.

Where lrPCR and Mito-SiPE widely diverged in results was in the mutation spectrum of wild-type C57BL6 mice. Long-range PCR amplification causes an increase in mutations occurring at the 'T' nucleotide position, whereas no mutations are identified using Mito-SiPE. This is also in contrast to both groups with a $Polg^{D257A/wt}$ background. Due to the large number of mutations that were identified in $Polg^{D257A/D257A}$ mouse tissues (1000–7500), high-frequency variants (≥10% MAF) were selected for further sequence analysis (Supplementary Fig. 10b). There was no difference between lrPCR and Mito-SiPE in the mutational spectrum in these high-frequency mutations in terms of profile or proportion of transitions to transversions. Additionally, we observed no difference in the loci/genes in which mutations were identified. The number of mutations found in each gene is available in Supplementary Data 1. The difference in heteroplasmic variants (MAF ≥0.2%) across the mitochondrial genome between both methodologies was calculated (Supplementary Fig. 11a). False positive heteroplasmic variants occur across the mitochondrial genome with a marked increase at the locations where lrPCR has reduced coverage and decrease in the D-loop region. Annotation of these differences did not show a notable contrast between low, medium and high-impact variants. However, more variants were identified in mt-Nd4 and mt-Nd1, which coincides with the lrPCR region of reduced coverage.

Finally, we attempted to use alternative methods of sequence-independent enrichment for human cell culture samples (Supplementary Fig. 12). Exonuclease digest, Qiagen's QProteome kit and QIAprep Miniprep kit were all used to isolate mitochondrial DNA from HepG2 cells. When the mtDNA copy number was assessed from these samples, a modest increase was observed. This increase is orders of magnitude lower than the

levels achieved using Mito-SiPE on fresh mouse tissue and would not be adequate as a substrate for ultra-deep sequencing.

## Discussion

In this study, we have demonstrated that a sequence-independent technique for mitochondrial DNA enrichment is highly effective and can produce ultra-deep sequencing coverage required for heteroplasmy analysis. We conclude that this methodology works most effectively with brain, heart, liver and kidney samples, however, sufficient results can also be obtained in samples originating from the lung, muscle and spleen. The cause of the disparity between tissues is unknown, however, we hypothesise that it may be related to the amount of starting material, the mechanical properties of the tissues and the mitochondrial copy number before enrichment. There are three main advantages of this method over the current standard: (1) It does not require complementary binding of reference probes/primers to mitochondrial DNA, (2) PCR amplification is not required and therefore generates no polymerase errors during enrichment and (3) Any nuclear contamination present is randomly distributed across the nuclear genome and therefore does not result in NUMT enrichment.

The sequencing depth achieved using this methodology exceeds that of many heteroplasmy studies to-date, even in the tissues where enrichment was less effective[5,19,38,39]. High coverage of the mitochondrial genome allows for a more sensitive assessment of low-frequency mutations and changes in heteroplasmy frequency. The number of reads (average 22.4 million per sample, Table 1) produced through sequencing in this study could be reduced by up to fivefold and would still be sufficient to achieve coverage levels in-line with previous studies (typically less than 10,000X). This would reduce the sequencing costs, increase throughput and enable the study of mitochondrial heteroplasmy across many samples by taking advantage of high-throughput sequencing. Our results also suggest that sufficient sequencing coverage could be achieved even with lower-capacity sequencing platforms.

An alternative bioinformatics pipeline is required when this technique is utilised. Typically, it is advised to align all sequencing data to the whole reference genome before isolating mitochondrial reads to avoid spurious alignment of nuclear reads to the mitochondrial genome[40]. However, due to the lack of sequence-specific enrichment when using this technique, a different approach is optimal. When reads are aligned to the whole reference genome first, mitochondrial reads are incorrectly aligned to homologous regions in the nuclear genome, most prominently, mouse chromosome 1. This effect in this study, it should be noted, is specific to the mouse reference genome; however, future studies may assess whether the same effect is observed in human alignments or, indeed, in other species. As we demonstrate here, mapping reads to the mitochondrial genome first did not appear to lead to an increase in the spurious alignment of nuclear reads to the mitochondrial genome and produced uniform sequencing coverage.

Nuclear-mitochondrial sequences have been identified as an important source of artefacts in heteroplasmy analysis. By employing sequence-independent enrichment of mitochondrial DNA, we find that any nuclear contamination present in the resultant sequence data is randomly distributed across the nuclear genome. It is pertinent to note, however, that although NUMT contamination is not enriched in this study, it does not mean that NUMT contamination is entirely absent. This is an important point as the number and size of all NUMTs have not been fully elucidated for most species and varies within species, even between individuals. The effect of NUMT contamination when using this technique will be minimised in the tissues that contain less nuclear contamination e.g., the brain, heart, kidney and liver.

Mitochondrial DNA heteroplasmy has been assessed in different ways across a number of studies. The number of heteroplasmic sites, average alternative allele frequency and cumulative heteroplasmic burden are all metrics that have been considered[4,14,34–37]. The number of heteroplasmic sites and cumulative heteroplasmic burden was higher in $Polg^{D257A/D257A}$ mouse tissues than in that of $Polg^{wt/wt}$, using both lrPCR amplification and mtDNA preparation methods. These findings are in-line with previous studies, however, the number of mutations detected appear to be higher in the data presented here. This may be due to the high levels of coverage achieved and a lower minimum threshold of alternative allele frequency (0.2%) utilised in this study. Interestingly, $Polg^{wt/wt}$ mice displayed lower average alternative allele frequencies than $Polg^{D257A/D257A}$. This effect, at first observation, is in stark contrast to what has been recorded previously and is not what one would expect based on the literature. However, the high number of low-frequency heteroplasmies identified in $Polg^{D257A/D257A}$ tissues have caused a decrease in the overall average heteroplasmy levels (average alternative allele frequency) compared to $Polg^{wt/wt}$ as a result of the high sequencing depth achieved across all samples. This result highlights the importance of looking at multiple measures of heteroplasmy to get an accurate assessment of the levels of mitochondrial DNA mutation that are present in a sample.

Mitochondrial DNA enriched via lrPCR amplification had a higher number of heteroplasmic sites and cumulative heteroplasmic burden than DNA enriched using the mtDNA prep method. This difference was larger in $Polg^{D257A/D257A}$ mice than in $Polg^{wt/wt}$. There are two likely explanations for this observation. First, that rounds of PCR amplification cause PCR errors that are subsequently identified as mitochondrial heteroplasmy, although a high-fidelity polymerase is used to negate this impact as much as possible. The second mechanism is that NUMT regions in the nuclear genome are being co-amplified and thus are mistaken for heteroplasmic, mitochondrial reads. If the first mechanism was the leading cause, it is unexpected that the difference would be higher in $Polg^{D257A/D257A}$ tissues than in $Polg^{wt/wt}$, as the error rate of the polymerase used for amplification should remain constant. Co-amplification of NUMT regions, however, could be affected by changes in mtDNA copy number – a feature that has previously been identified in $Polg$ mutator mice[41–43]. Changes in mtDNA copy number may directly affect the amount of mispriming of NUMTs that occurs during lrPCR. Whilst the results described here cannot rule out either mechanism, this evidence suggests that the co-amplification of NUMT regions is a more likely/influential candidate mechanism to explain the false positive heteroplasmic variants identified in PCR amplified samples. The metric of average alternative allele frequency did not display this same pattern, but as explained previously, it is influenced by the presence of many apparent, low-frequency heteroplasmies that are identified when using lrPCR amplification at these sequencing depths and heteroplasmy thresholds. This effect was even more dramatic when lrPCR was compared to mtDNA enrichment prep of C57BL6 wild-type tissues. This is largely due to the high baseline of mutation that is present in $Polg^{wt/wt}$ due to the intermediate phenotype of female breeder mice.

Analysis of the variants identified using lrPCR and Mito-SiPE showed reproducible results across $Polg^{wt/wt}$ and $Polg^{D257A/D257A}$ mouse tissues. Mito-SiPE showed much improvement in samples that have low levels of mitochondrial heteroplasmy. Long-range PCR amplification causes artificially elevated levels of mutation, either through PCR errors or amplification of NUMTs. Mito-SiPE offers researchers a more sensitive approach to detect smaller

changes in low-frequency mutations than methods reliant on PCR.

Finally, a tissue-specific effect on the levels of heteroplasmy was observed using the mtDNA prep method in $Polg^{D257A/D257A}$ mice. Spleen and colon samples had higher levels of heteroplasmy compared to heart, kidney, liver and lung, whereas, brain samples had lower levels. Heart, kidney, liver and lung were indistinguishable from one another. This is an interesting finding, as previous studies have identified strong, tissue-specific effects of mitochondrial heteroplasmy in more tissues than reported here[4,5,44]. It is possible that enrichment methods used in previous studies have identified false heteroplasmic variants due to NUMT co-enrichment, and thus the differences observed are due to mtDNA copy number changes, rather than true mtDNA mutations. It is also possible, however, that the levels of coverage and low thresholds used in this study have led to this difference.

One limitation of this methodology is that it requires the availability of tissue for mitochondrial enrichment, and as such, it may not be feasible for archived DNA samples. DNA purified from intact cells and tissues using standard methods predominantly consists of nuclear DNA. These preparations and already existing NGS data are not compatible with this method. We attempted to use alternative methods of mtDNA enrichment, such as exonuclease digest[32], however, the level of enrichment and total yield of DNA would not be compatible with ultra-deep sequencing of mitochondrial DNA.

With the explosion of interest in mtDNA and increasing levels of heteroplasmy-focused research, Mito-SiPE will provide an important tool for future explorations of mtDNA variation and its role in aging and disease. Our methodology is limited by the sequencing error rate that produces false positive heteroplasmy calls. The use of unique molecular identifiers (UMIs) have been shown to reduce the impact of errors and thus increase one's ability to detect rare variants[45,46]. UMI's are nucleotide 'barcodes' which are ligated to DNA before library preparation. After sequencing, consensus reads are generated from reads that possess the same UMI, and thus PCR errors and sequencing errors can be negated. Mito-SiPE, in combination with the use of UMIs, may enable researchers to detect variants at even lower levels than what is documented here.

In conclusion, differential centrifugation and alkaline lysis may be used to enrich mitochondrial DNA free from PCR amplification or probe hybridisation. Avoiding sequence-dependent techniques greatly reduces the effect of NUMT contamination, a problem which has been identified in the previous studies[23,24,28,30]. This technique, in addition to a modified bioinformatics pipeline, can be applied to different tissues and achieves ultra-deep sequencing coverage. It provides a straightforward and robust workflow to assess heteroplasmy in mitochondrial DNA. This technique outperforms long-range PCR amplification and negates the potential impact of PCR errors and NUMT contamination on heteroplasmy analysis.

## Methods

**Breeding and tissue harvesting**. Mice were housed in shoe box cages and fed ProLab RMH 1800 diet (PMI Nutrition International) containing 50 μg vitamin B12/kg of diet and 3.3 mg folic acid/kg of diet. Breeding mice were fed Picolab Mouse Diet 20, containing 51 μg vitamin B12/kg diet and 2.9 mg folic acid/kg of diet. Heterozygous $Polg^{wt/D257A}$ males and $Polg^{wt/D257A}$ females were mated. Homozygous mutant and wild-type progeny were aged to 6 months, at which point they were sacrificed. There were 4 $Polg^{wt/wt}$ mice (two male and two female) and 4 $Polg^{D257A/D257A}$ mice (two male and two female) used in the $Polg$ experiments. Brain, heart, lung, liver, spleen, kidney and muscle tissue was isolated and mitochondrial DNA enrichment was performed on all tissues.

**Tissue homogenisation**. Harvested tissue was placed in a homogenisation tube with 10X volume per gram of fresh homogenisation buffer i.e. 5 ml buffer for 500 mg tissue. Tissues were homogenised until no discernible whole tissue was

present. The homogenate was then transferred to 1.5 ml microcentrifuge tubes and spun at $1000 \times g$ for 1 min at 4 °C. The supernatant was transferred to a new microcentrifuge tube and spun at $12,000 \times g$ for 10 min at 4 °C to pellet mitochondria. The mitochondrial pellet was resuspended with 100 μl of resuspension buffer for storage or for immediate DNA extraction.

**Mitochondrial DNA isolation from fresh mouse tissue**. The mitochondria resuspension was added to 200 μl alkaline lysis buffer, vortexed, and placed on ice for 5 min. Potassium Acetate Buffer (150 μl) was then added and the mixture was vortexed slowly and placed on ice for 5 min. The mixture was centrifuged at $12,000 \times g$ for 5 min at 4 °C to pellet proteins and the supernatant was decanted to a new tube. RNase (1 μg) was added to the mixture and left at room temperature for 15 min. Phenol-chloroform (500 μl) was added to each tube, inverted and placed on a shaker/rotator for 20 min. Afterwards, centrifugation at $12,000 \times g$ for 2 min at room temperature was carried out. The aqueous (top) layer was decanted to a new tube (~450 μl from this phase was retrieved) and 40 ul sodium acetate, 1 μl glycogen (20 mg/ml) and 1200 μl 100% EtOH were added. The mixture was inverted and mixed well, then left on dry ice for 60 min. The mixture underwent centrifugation at $12,000 \times g$ and the supernatant was removed. The pellet was finally washed twice using 70% ethanol, air-dried, and resuspended in a low-TE buffer for sequencing or regular TE buffer for (q)PCR.

**Library preparation and next-generation DNA sequencing**. Libraries were generated from approximately 50 ng genomic DNA using the Accel-NGS 2 S Plus DNA Library Kit (Swift Biosciences) using five cycles of PCR to minimise PCR bias. The DNA samples were sheared by sonication (Covaris Inc., Woburn, MA) to a mean of 300 bp. Libraries were tagged with unique dual index DNA barcodes to allow the pooling of libraries and minimise the impact of barcode hopping. Libraries were pooled for sequencing on the NovaSeq 6000 (Illumina) to obtain at least 7.6 million 151-base read pairs per individual library. Sequencing data was processed using RTA version 3.4.4. DNA sequencing was carried out at the NIH Intramural Sequencing Center.

**Data processing and alignment**. Fastq files were aligned to the mouse reference genome, GRCm38, using bwa mem using the default parameters[47]. Picard tools were used to add read groups, and to mark and remove duplicates[48]. Samtools was used to calculate the coverage across the nuclear and mitochondrial genome for each sample. Finally, R (v3.5.0) and ggplot2 (v3.3.0) were used for statistical analysis and subsequent visualisation of graphs[49,50]. Library complexity and fragment sizes were calculated using Picard tools v1.4.2 on 15 randomly-selected samples (Supplementary Data 2).

**Variant calling and mutation analysis**. Variant calling was performed using bcftools v1.9 with 'bcftools mpileup -f -Q 30 –skip-indels reference_fasta bam_file | bcftools call -mv' to identify single nucleotide variants only. Filtering was performed by removing any SNVs that had a QUAL score lower than 20. The code used for alignment and variant calling is available on github (https://github.com/walshd59/mtDNAhetScripts.git). Of 66,738 variants identified across all samples in our study at an alternative allele frequency ≥0.2%, only 137 of these had an ln(Strand Odds Ratio) value ≥3. Analysis of the mutation spectrum and further characterisation/annotation of heteroplasmic variants was performed using SnpEff (v 5.1)[51].

**Quantification of mtDNA copy number**. Mitochondrial DNA copy number was assessed via qPCR targeting both mitochondrial and nuclear loci, as previously described[52,53]. Briefly, 2.5 μl LightCycler® 480 SYBR Green I Master (Roche, Molecular Systems, Inc, Germany), 2 μl of DNA (20 ng/μl) and 0.5 μl primer mix were added in triplicate to a 384-well plate and the reactions were carried out by the QuanStudio 6 Flex (Applied Biosystems, Foster City, CA, USA). The conditions were as follows: 95 °C for 5 min, 45 cycles of 95 °C for 10 s, 60 °C for 10 s and 72 °C for 20 s. A melting curve was performed using 95 °C for 5 s, 66 °C for 1 min and gradual increase to 97 °C. The mitochondrial DNA copy number was assessed using the following equation:

$$2 \times 2^{\Delta Ct} (\text{where } \Delta Ct = Ct(\text{mtDNA gene}) - Ct(\text{nDNA gene})) \qquad (1)$$

The following primers were used for human mtDNA copy number: mtDNA tRNA (forward: CACCCAAGAACAGGGTTTGT, reverse: TGGCCATGGGTATG TTGTTA) nuclear DNA β2-microglobulin (forward: TGCTGTGTCTCCATGTTT GATGTATCT), reverse: TCTCTGCTCCCCACCTCTAAGT). Mouse mtDNA copy number was assessed using the following primers: mtDNA ND1 (forward: CTAGCAGAAACAAACCGGGC, reverse: CCGGCTGCGTATTCTACGTT) nuclear DNA HK2 (forward: GCCAGCCTCTCCTGATTTTAGTGT, reverse: GGGAACACAAAAGACCTCTTCGG). These primers are available in the supplementary data file (Supplementary data 3).

**Long-range PCR enrichment of mtDNA**. This technique was used to amplify human and mouse mitochondrial DNA in two fragments from a whole DNA extract. DNA was quantified via Nanodrop (Methods 2.2.7) unless otherwise

stated. Each PCR reaction consisted of Q5 High-fidelity Polymerase (0.02 U/µl), 5X Q5 reaction buffer (1X), 10 mM dNTPs (300 µM), 5 µM Forward and Reverse primers (0.25 µM). Template DNA (100 ng) was added to each reaction except for the no-template control (NTC) but an equivalent volume of molecular biology-grade water was added instead. The temperature cycles were as follows: $1 \times 30$ s denature 98 °C, $25 \times 10$ s denature 98 °C, 30 s annealing 66 °C, 4 min 30 s elongation 72 °C, $1 \times 10$ min elongation 72 °C on a thermocycler. Both fragments were quantified using Qubit and mixed in equimolar ratios. The following primers were used for each fragment: lrPCR fragment 1 (forward: GGATCCTACTCTCTA CAAAC, reverse: TAGTTTGCCGCGTTGGGTGG) and lrPCR fragment 2 (forward: CTACCCCCTTCAATCAATCT, reverse: CCGGTTTGTTTCTGCTAGGG). These primers are also available in the supplementary data file (Supplementary Data 3).

**Mitochondrial isolation via Qiagen QProteome™ kit**. HepG2 cells ($2 \times 10^6$) were collected, counted and pelleted when 80% confluency was reached. The supernatant was removed and the pellet was resuspended in the lysis buffer provided in the kit. Homogenisation and mitochondrial isolation were then carried out as per the manufacturers' protocol. Briefly, the cell pellet was resuspended in 1.5 ml of ice-cold Disruption Buffer by pipetting up and down using a 1 ml pipet tip. The cell disruption was completed using a blunt-ended needle and a syringe. The mixture was centrifuged at $1000 \times g$ for 10 min at 4 °C to pellet proteins and the supernatant was decanted to a new tube. Afterwards, centrifugation at $6000 \times g$ for 2 min at room temperature was carried out. This pellet was then resuspended in 200 µl PBS and 20 µl proteinase K. DNA extraction was then performed on the mitochondrial isolate via Qiagen DNeasy™ Blood and Tissue kit using the manufacturer's protocol.

**Plasmid-Safe™ digest for mtDNA enrichment**. Whole DNA extractions were treated with Plasmid-Safe ATP-dependent exonuclease (Lucigen) as per the manufacturer's protocol. Briefly, a plasmid-safe solution was created using 42 µl sterile water, 2 µl 25 mM ATP, 5 µl 10X reaction buffer and 1 µl Plasmid-Safe DNase. This DNase targets linear molecules and, as such, does not degrade intact mitochondrial DNA as it is circular. The solution was added to the DNA extracted using the QIAprep miniprep and incubated at 37 °C for 1 h. The DNase was then deactivated with a 70 °C incubation for 30 min.

**HepG2 culture conditions**. HepG2 cells (Merck; 85011430-1VL) were cultured in Dulbecco's modified Eagle medium (DMEM) with 10% supplementation of foetal bovine serum in a 5% CO$_2$ incubator. Cells were cultured in 10 ml of media in T75 flasks. The cells were passaged by washing with 5 ml PBS (1X) followed by incubation in 2 ml of 0.25% Trypsin-EDTA at 37 °C for 5 min. Trypsinisation was inhibited by adding 4 ml DMEM. Cells were then collected via centrifugation at $500 \times g$ for 5 min before being counted.

**Qiagen QIAprep miniprep**. HepG2 cells ($2 \times 10^6$) were collected, counted and pelleted when 80% confluency was reached. The supernatant was removed and the pellet was resuspended in lysis buffer provided in the kit. DNA isolation was performed as per the manufacturer's protocol, using a silica membrane to capture the mtDNA, which was subsequently collected in 100 µl the provided elution buffer.

**Statistics and reproducibility**. All statistical analyses included in this paper were carried out in R (version 4.1.1) and the software package rstatix (version 0.7.0). Sample sizes are described within each experimental figure. Wilcoxon signed-rank tests were performed to compare mtDNA copy number from unenriched and enriched samples from different tissues ($n = 14$ for each group). Wilcoxon signed-rank tests were also performed to compare the effect of lrPCR amplification and Mito-SiPE on mitochondrial heteroplasmy in $Polg^{wt/wt}$ and $Polg^{D257A/D257A}$ ($n = 24$ for each genotype). A Student's $t$-test was used for comparing the effect of lrPCR and Mito-SiPE on heteroplasmy in C57BL6 wild-type mice ($n = 6$ for the lrPCR group, $n = 12$ for the Mito-SiPE group).

**Solutions**. The following solutions were used: Homogenisation Buffer (0.25 M Sucrose, 10 mM EDTA, 30 mM Tris-HCl, pH = 7.5), Resuspension Buffer (10 mM Tris, 0.15 M NaCl, 10 mM EDTA, pH = 8.0), Alkaline Lysis Buffer (0.18 N NaOH, 1% SDS, prepared fresh), Potassium Acetate Buffer (3 M potassium, 5 M acetate), and Low-TE buffer (10 mM Tris-HCl, 0.1 mM EDTA, pH = 8.0).

**Reporting summary**. Further information on research design is available in the Nature Portfolio Reporting Summary linked to this article.

## Data availability
All sequencing data and associated metadata is available on Sequence Read Archive (PRJNA881035). The raw fastq files may be downloaded from this repository using prefetch or similar https://www.ncbi.nlm.nih.gov/bioproject/PRJNA881035. All source data used to make graphs are included in the supplementary data file (Supplementary Data 4).

## Code availability
The code used for alignment and variant calling is available on GitHub (DOI: 10.5281/zenodo.7238550).

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

## Acknowledgements

This work was funded by the Wellcome Trust, the Intramural Program of the National Human Genome Research Institute and the NIH Intramural Sequencing Center.

## Author contributions

D.J.W., M.E. and D.J.B. performed the sample collections and enrichment. D.J.W. and D.H. performed bioinformatics and subsequent analysis. D.J.W., D.J.B., F.P., D.H., A.P.McD. and L.C.B. prepared the manuscript. All authors read and approved the final manuscript.

## Funding

## Competing interests

The authors declare no competing interests.

## Ethical approval

All animal protocols were reviewed and approved by the National Human Genome Research Institute (NHGRI) Animal Care and Use Committee prior to animal experiments.
