## [Peer Review File · Communications Biology]

Reviewers' comments:

Reviewer #1 (Remarks to the Author):

Walsh et al present Mito-SiPE, a new method that enriches for mitochondria before library prep to derive extremely high quality mtDNA for rare heteroplasmy analysis. The method is well justified and the presentation looks technically impressive. The application to the PolG mutator mice is appropriate, and the results clearly indicate that Mito-SiPE can be a valuable tool in the mtDNA analysis toolkit. Overall, the manuscript is in good shape and I commend the authors for their work. I have a few suggestions that may improve the work:

- As far as I can tell, there is no explicit variant calling performed. How do the authors recommend identifying variants from Mito-SiPE data? Sequencing errors, as an example, tend to be non-random where certain regions are more likely than others to yield false heteroplasmy due to Illumina sequencers. I'd recommend a very minimum verifying that the strand concordance (measured by a Fishers test of ref/alt allele and +/- strand) be assessed for variant filtering.

- Previously, when ultra rare sequencing method is developed, a UMI is used to further mitigate PCR bias, perhaps using a consensus collapsing (e.g. see: <https://www.ncbi.nlm.nih.gov/pmc/articles/PMC6723130/>). If the authors could comment on this approach relative to a UMI-based method, this could help presentation.

- Additional analyses of the heteroplasmies, such as annotation in specific genes, complex families, synonymous/ nonsynonymous/ etc., would improve the demonstration of why one should sequence with this approach (i.e. can the authors better justify, using their data, why to use their approach in order to learn new biology).

- It would be of interest to share the distribution of fragment sizes and the percent duplicates (both metrics easily derived from Picard tools) for these libraries or for a subset of them.

- I would encourage the authors to share sequencing data on GEO or a related means as the method is much more likely to be adopted if the data is easily and freely accessible.

Reviewer #2 (Remarks to the Author):

In this manuscript, Walsh et al. describe Mito-SiPE, a method for extracting and enriching Mitochondrial DNA without relying on PCR or probes. The authors showed that most reads in the generated libraries map the mitochondrial genome. Then the authors showed that nuclear-mitochondrial sequences (NUMT) do not contribute significantly to the higher levels of mtDNA mapped reads. Moreover, the authors demonstrated that their technique generates more uniform coverage across the mitochondrial genome than PCR-based methods. Finally, the authors showed that Mito-SiPE significantly reduces false positive variants.

Overall, the technique will interest readers, especially those studying mtDNA mutations. However, it is not clear what is the manuscript's advantage over what Gould et al. demonstrated in 2015 (PCR-Free Enrichment of Mitochondrial DNA from Human Blood and Cell Lines for High Quality Next-Generation DNA Sequencing, Plos One 2015). I have the following concerns that the authors should address before determining whether Mito-SiPE indeed presents an advantage over other techniques:

1- The authors should discuss Gould et al., 2015 and highlight the advantage of Mito-SiPE over that study (if advantages are present).

2- In Figure 2b (top), the authors claim that the lost coverage in the middle part is due to NUMT and

showed that if they map to mtDNA, the coverage is similar to surrounding regions. I recommend demonstrating the pair-wise alignment between that region in mitochondria and the gDNA to substantiate this.

3- Figure 2C showed that gDNA-mapped reads have higher enrichment in Chromosomes 2 and 9. According to the authors, the alignment tool assigned highly repetitive reads to these two chromosomes. The authors used bwa for mapping, which is known to cause this issue. Removing such reads from the bam file before studying the distribution is highly recommended. Use another mapper that can remove multi-mapped read. Alternatively, such reads will have the Tag "XA:Z" or "SA:Z" in the bam file generated by BWA. You can remove reads with such tags using various tools.

4- Supplementary figures 2-5 are not clear. Add a key to the phylogenetic trees to understand the similarity rate.

5- The comparison in terms of mapped reads between IrPCR and Mito-SiPE should be made in percentages to appreciate the difference in efficiency. Mito-SiPE seems to be having significantly lower mapping percentage than IrPCR.

Minor Comment:

What is the full Mito-SiPE? (Sequence-independent PCR-Free Enrichment)? It is nowhere in the main text.

Reviewer #1 (Remarks to the Author):

Walsh et al present Mito-SiPE, a new method that enriches for mitochondria before library prep to derive extremely high quality mtDNA for rare heteroplasmy analysis. The method is well justified and the presentation looks technically impressive. The application to the PolG mutator mice is appropriate, and the results clearly indicate that Mito-SiPE can be a valuable tool in the mtDNA analysis toolkit. Overall, the manuscript is in good shape and I commend the authors for their work. I have a few suggestions that may improve the work:

1) As far as I can tell, there is no explicit variant calling performed. How do the authors recommend identifying variants from Mito-SiPE data? Sequencing errors, as an example, tend to be non-random where certain regions are more likely than others to yield false heteroplasmy due to Illumina sequencers. I'd recommend a very minimum verifying that the strand concordance (measured by a Fishers test of ref/alt allele and +/- strand) be assessed for variant filtering.

This remark is much appreciated as typically a specific variant caller is used to identify rare mutations present in the mitochondrial genome or somatic mutations in tissue/cancer samples. The variant calling in our analysis was carried out using bcftools to count occurrences of reference and alternative alleles after duplication removal. We have altered the methods to make this clearer in the text. During this process both duplication and base quality score were all considered. Whilst we did not specifically filter for strand odds ratio or Fisher's values, our existing variant calling strategy and frequency threshold essentially eliminated any strand biases present. Of 66,738 variants identified across all samples in our study at an alternative allele frequency $\geq 0.2\%$, only 137 of these had a $\ln(\text{Strand Odds Ratio})$ value ≥ 3 . This represents a negligible fraction of total variants observed and omission of these variants had no impact on final analysis. Additionally, we have included a github repository with the code used for alignment and variant calling in the methods section.

We have modified the text in the manuscript at Lines 657-666:

Variant calling and mutation analysis

Variant calling was performed using bcftools v1.9 with 'bcftools mpileup -f -Q 30 --skip-indels reference_fasta bam_file | bcftools call -mv' to identify single nucleotide variants only. Filtering was performed by removing any SNVs that had a QUAL score lower than 30. The code used for alignment and variant calling is available on github (<https://github.com/walshd59/mtDNAhetScripts.git>). Of 66,738 variants identified across all samples in our study at an alternative allele frequency $\geq 0.2\%$, only 137 of these had a $\ln(\text{Strand Odds Ratio})$ value ≥ 3 .

2) Previously, when ultra rare sequencing method is developed, a UMI is used to further mitigate PCR bias, perhaps using a consensus collapsing (e.g. see: <https://www.ncbi.nlm.nih.gov/pmc/articles/PMC6723130/>). If the authors could comment on this approach relative to a UMI-based method, this could help presentation.

The utilisation of UMIs to increase sensitivity of the methodology described is an excellent suggestion and something we anticipate testing in the future. Although our method currently represents a powerful tool for rare-heteroplasmy identification, using a UMI may further enhance its performance. We have included a short comment in the discussion which addresses this point.

Lines 419-427 have been modified in the revised manuscript.

Our methodology is limited by the sequencing error rate that produces false positive heteroplasmy calls. The use of unique molecular identifiers (UMIs) have been shown to reduce the impact of errors and thus increase one's ability to detect rare variants⁴⁶. UMI's are nucleotide 'barcodes' which are ligated to DNA before library preparation. After sequencing, consensus reads are generated from reads that possess the same UMI and thus PCR errors and sequencing errors can be negated. Mito-SiPE, in combination with the use of UMIs, may enable researchers to detect variants at even lower levels than what is documented here.

3) *Additional analyses of the heteroplasmies, such as annotation in specific genes, complex families, synonymous/ nonsynonymous/ etc., would improve the demonstration of why one should sequence with this approach (i.e. can the authors better justify, using their data, why to use their approach in order to learn new biology).*

Thank you for this suggestion. As part of our revised manuscript we have included an analysis of the variants identified in POLG and wild-type mice. Unsurprisingly, Mito-SiPE and IrPCR show very similar results in samples that have high levels of mitochondrial mutation. However, Mito-SiPE negates the impact of PCR errors and artifact-generating NUMT amplification which is evident in the lack of mutations found in C57BL6 mice. In this way, it offers researchers a reliable method of low-frequency variant identification and negates the impact of false positives driven by IrPCR or similar methods. There was no significant difference between Mito-SiPE and IrPCR in terms of location/genes in which mutations were found. Additionally, we calculated the difference in heteroplasmic variants (MAF \geq 0.2%) across the mitochondrial genome between both methodologies (Supplementary Fig. 11, a). This figure shows that false positives occur almost across the whole mitochondrial genome with a marked increase at the locations where IrPCR has reduced coverage. Annotation of these differences did not show a notable difference between low, medium and high impact variants. However more variants were identified in mt-Nd4 and mt-Nd1, which coincides with the IrPCR region of reduced coverage.

The revised manuscript has been modified at Lines: 266-292:

Analysis of the mutation profile observed using both IrPCR and Mito-SiPE showed similar results in $Polg^{wt/wt}$ and $Polg^{D257A/D257A}$ (Supplementary Fig 10a). $Polg^{D257A/D257A}$ mice had higher levels of mutations occurring at 'C' nucleotide positions in the reference genome (light-strand). Interestingly, $Polg^{wt/wt}$ had more mutations at the 'A' nucleotide position, indicating that perhaps there is some selection that occurs when mutations are passed from $Polg^{D257A/wt}$ to their progeny.

Where IrPCR and Mito-SiPE widely diverged in results was in the mutation spectrum of wildtype C57BL6 mice. Long-range PCR amplification causes an increase in mutations occurring at the 'T' nucleotide position whereas no mutations are identified using Mito-SiPE. This is also in contrast to both groups with a $Polg^{D257A/wt}$ background. Due to the large number of mutations that were identified in $Polg^{D257A/D257A}$ mouse tissues (1000-7500), high-frequency variants (\geq 10% MAF) were selected for further

sequence analysis (Supplementary Fig 10b; Supplementary data). There was no difference between IrPCR and Mito-SiPE in the mutational spectrum in these high-frequency mutations in terms of profile or proportion of transitions to transversions. Additionally, we observed no difference in the loci/genes in which mutations were identified. The number of mutations found in each gene is now available in Supplementary Data, Table 1. The difference in heteroplasmic variants (MAF \geq 0.2%) across the mitochondrial genome between both methodologies was calculated (Supplementary Fig. 11, a). False positive heteroplasmic variants occur almost across the mitochondrial whole genome with a marked increase at the locations where IrPCR has reduced coverage and decrease in the D-loop region. Annotation of these differences did not show a notable contrast between low, medium and high impact variants. However, more variants were identified in mt-Nd4 and mt-Nd1, which coincides with the IrPCR region of reduced coverage.

The revised manuscript has been modified at Lines 393-399:

Analysis of the variants identified using IrPCR and Mito-SiPE showed reproducible results across Polgwt/wt and PolgD257A/D257A mouse tissues. Mito-SiPE showed improvement in samples that have low to no levels of mitochondrial heteroplasmy. Long-range PCR amplification causes artificially elevated levels of mutation, either through PCR errors or amplification of NUMTs. Mito-SiPE offers researchers a more sensitive approach to detect smaller changes in low-frequency mutations compared to methods that are reliant on PCR.

4) *It would be of interest to share the distribution of fragment sizes and the percent duplicates (both metrics easily derived from Picard tools) for these libraries or for a subset of them.*

Thank you for this suggestion. We have included tables for the library complexity and fragment size metrics as calculated by Picard tools in Supplementary Table 2. These have been calculated for a subset of samples (n = 15) randomly selected from the library.

Library complexity metrics (Picard tools; mean values)		
Optical duplicates (sd)	Percent duplication (sd)	Library size (sd)
702,000 (110,000)	0.52 (0.12)	23,000,000 (4,700,000)

Fragment size metrics (Picard tools; mean values (sd))	
MEDIAN_INSERT_SIZE	346 (18)
MODE_INSERT_SIZE	209 (13)
MEDIAN_ABSOLUTE_DEVIATION	116 (10)
MIN_INSERT_SIZE	3 (2)
MAX_INSERT_SIZE	16297 (6)
MEAN_INSERT_SIZE	365 (18)
STANDARD_DEVIATION	160 (11)
READ_PAIRS	6317968 (1628243)
WIDTH_OF_10_PERCENT	46 (4)
WIDTH_OF_20_PERCENT	92 (8)

WIDTH_OF_30_PERCENT	139 (12)
WIDTH_OF_40_PERCENT	186 (16)
WIDTH_OF_50_PERCENT	234 (21)
WIDTH_OF_60_PERCENT	284 (25)
WIDTH_OF_70_PERCENT	336 (29)
WIDTH_OF_80_PERCENT	394 (32)
WIDTH_OF_90_PERCENT	497 (35)
WIDTH_OF_95_PERCENT	634 (45)
WIDTH_OF_99_PERCENT	1065 (522)

Lines 654-655:

Library complexity and fragment sizes were calculated using Picard tools v 1.4.2 on 15 randomly-selected samples (Supplementary Table 2).

5) *I would encourage the authors to share sequencing data on GEO or a related means as the method is much more likely to be adopted if the data is easily and freely accessible.*

We appreciate this comment and have submitted our sequences to NCBI's SRA. We have uploaded all of the raw reads under the submission reference: SUB12051683 (BioProject: PRJNA881035).

Reviewer #2 (Remarks to the Author):

In this manuscript, Walsh et al. describe Mito-SiPE, a method for extracting and enriching Mitochondrial DNA without relying on PCR or probes. The authors showed that most reads in the generated libraries map the mitochondrial genome. Then the authors showed that nuclear-mitochondrial sequences (NUMT) do not contribute significantly to the higher levels of mtDNA mapped reads. Moreover, the authors demonstrated that their technique generates more uniform coverage across the mitochondrial genome than PCR-based methods. Finally, the authors showed that Mito-SiPE significantly reduces false positive variants.

Overall, the technique will interest readers, especially those studying mtDNA mutations. However, it is not clear what is the manuscript's advantage over what Gould et al. demonstrated in 2015 (PCR-Free Enrichment of Mitochondrial DNA from Human Blood and Cell Lines for High Quality Next-Generation DNA Sequencing, Plos One 2015). I have the following concerns that the authors should address before determining whether Mito-SiPE indeed presents an advantage over other techniques:

1- The authors should discuss Gould et al., 2015 and highlight the advantage of Mito-SiPE over that study (if advantages are present).

Thank you for this very insightful suggestion. It was an oversight on our behalf to not include this paper in the introduction and we have corrected that in the revised submission. In unpublished work, we have attempted to use exonuclease digests to enrich mitochondrial DNA. We then quantified the level of enrichment using a qPCR

assay, similarly to Gould et al. 2015. Our qPCR assay results were broadly similar to Gould et al. except we could not achieve efficiency levels as high as their best sample (61.78% of reads aligned to the mitochondrial genome). More importantly, the amount of starting material and/or final DNA concentration post-digest, in our hands, was a substantial limiting factor in the amount of reads that would be achievable and therefore cause significant (1000-to-10,000-fold) reduction in the per-base coverage of the mitochondrial genome. In essence, it may not always be feasible to use this methodology to achieve ultra-high sequencing depth of the mitochondrial genome and thus, rare heteroplasmy analysis. The following is an excerpt from the thesis (from DJW) to be published in the coming months that featured the aforementioned work:

“The exonuclease digest enrichment method results were also comparable to previously published literature (Gould et al. 2015). This group first measured the relative mtDNA levels of blood that was treated with Plasmid-Safe exonuclease using a qPCR assay, similar to the one used in this project. Whilst their qPCR results were comparable to those displayed above, they also performed next-generation sequencing on the same samples and found that a maximum of 61.78% of reads aligned to the mitochondrial genome for one of their samples. This is much higher than our estimation but may be explained by the small total read count achieved during sequencing of said sample (total reads aligned = 16,156) (Gould et al., 2015).”

This figure shows results from said unpublished thesis that compared mtDNA copy number post-enrichment using exonuclease digest, a miniprep kit and Qproteome isolation followed by DNA extraction. These enrichments were carried out on HepG2 cells which may explain some of the discrepancies between our work and Gould et al., who used human blood, the metastatic melanoma cell line COLO 829 and its matched normal lymphoblast line COLO 829BL. We commend Gould et al.’s work as it offers researchers a sequence-independent method that works on cell culture, which is a major difference to what we’re reporting here. However, we feel that the final mtDNA read depth and variety of tissues reported in our paper offers a substantial difference/improvement to their method. Finally and importantly, the small number of

aligned reads achieved using Gould et al.'s method makes it unsuitable for low-frequency heteroplasmy quantification at the levels reported in this manuscript. lines 112-117:

A sequence independent method for mitochondrial enrichment carried out on blood and cell culture samples was previously described using an exonuclease digest⁴⁷. Its use on samples with modest amount of starting material offers a unique approach for measuring heteroplasmy in scarce samples such as cultured cells. However, compared to Mito-SiPE, the method yielded a reduced mapping efficiency and sequencing depth on average, limiting its use in low frequency heteroplasmy analysis.

2- In Figure 2b (top), the authors claim that the lost coverage in the middle part is due to NUMT and showed that if they map to mtDNA, the coverage is similar to surrounding regions. I recommend demonstrating the pair-wise alignment between that region in mitochondria and the gDNA to substantiate this.

Thank you for the suggestion. We have included a screenshot from BLAST below. When we query the mitochondrial sequence (MT:7500-11000) against the mouse genome a hit from Chromosome 1 appears and has a 99.94% sequence identity across the full length of the query. In our original text we erroneously claimed 100%, which has now been rectified. The figure below has been included as Supplementary Figure 2.

Description	Scientific Name	Max Score	Total Score	Query Cover	E value	Per. Ident	Acc. Len	Accession
Mus musculus strain C57BL/6J chromosome 1, GRCm39	Mus musculus	6455	7515	100%	0.0	99.94%	195154279	NC_000067.7

We have also modified Lines 181-183 in the revised manuscript. This is due to an NUMT found on chromosome 1 of the reference genome that shares 99.94% sequence identity of its homologous sequence in the mitochondrial genome (Supplementary Fig. 2).

3- Figure 2C showed that gDNA-mapped reads have higher enrichment in Chromosomes 2 and 9. According to the authors, the alignment tool assigned highly repetitive reads to these two chromosomes. The authors used bwa for mapping, which is known to cause this issue. Removing such reads from the bam file before studying the distribution is highly recommended. Use another mapper that can remove multi-mapped read. Alternatively, such reads will have the Tag "XA:Z" or "SA:Z" in the bam file generated by BWA. You can remove reads with such tags using various tools.

Thank you for this suggestion. We agree that if one were to look more closely at the distribution across each chromosome, that removal of these multi-mapped reads would be beneficial. However, our main aim from this demonstration was to show that real mitochondrial reads were being misaligned to Chromosome 1 due to a region with high levels of sequence identity. The repetitive reads that are aligned to Chromosomes 2 and 9 are trivial in this regard. When we removed the reads with the tags "XA:Z" or "SA:Z" in a subset of the samples, the observed elevated coverage was reduced but evident (shown below).

4- *Supplementary figures 2-5 are not clear. Add a key to the phylogenetic trees to understand the similarity rate.*

Thank you for this remark. We have replaced the phylogenetic trees with BLAST results tables to show that the regions on chromosome 2 and 9 have other regions in the mouse genome that share a high level of sequence identity and thus leads to misalignment at these loci.

5- *The comparison in terms of mapped reads between IrPCR and Mito-SiPE should be made in percentages to appreciate the difference in efficiency. Mito-SiPE seems to be having significantly lower mapping percentage than IrPCR.*

Thank you for this observation. We have edited Table 3 to include the percentage of reads aligned to the mitochondrial genome. Your statement is totally correct, IrPCR has a much higher mapping efficiency as has been documented in previous literature (including the Gould et al. paper). Our findings show that even with this lower mapping efficiency, Mito-SiPE increases one's ability to detect low-frequency mutations due to avoiding false-heteroplasmic calls.

Minor

Comment:

What is the full Mito-SiPE? (Sequence-independent PCR-Free Enrichment)? It is nowhere in the main text.

Thank you for this observation, we have defined Mito-SiPE at the end of the introduction.

See Lines 106-107 of the revised manuscript.

We have called this method Mito-SiPE (A sequence-independent, PCR-free mitochondrial DNA enrichment)

Mito-SiPE: A sequence-independent, PCR-free mitochondrial DNA enrichment
method for ultra-deep sequencing that minimises amplification and alignment artifacts.

Darren J Walsh^{1,2}, David J Bernard¹, Faith Pangilinan¹, Madison Esposito¹, Denise
Harold², Anne Parle-McDermott², Lawrence C Brody¹ (corresponding author).

1 - Gene and Environment Interaction Section, National Human Genome Research
Institute, NIH, Bethesda, MD, USA.

2 – School of Biotechnology, Dublin City University, Dublin, Ireland.

**Abstract**

**Background**

The analysis of somatic variation in the mitochondrial genome requires deep
sequencing of the mitochondrial genome. This is ordinarily achieved by selective
enrichment methods, such as PCR amplification or probe hybridization. These
methods can introduce bias and are prone to contamination by nuclear-mitochondrial
sequences (NUMTs), elements that can introduce artefacts into heteroplasmy
analysis.

**Results**

Here, we demonstrate a method to obtain ultra-deep (>80,000X) sequencing coverage
of the mitochondrial genome by selectively purifying the organelle with differential
centrifugation and alkaline lysis of seven different mouse tissues. This method yields
a preparation of highly enriched mtDNA and avoids the pitfalls inherent in the widely
employed sequence dependent methods. This methodology avoids false-
heteroplasmy calls that occur when long-range PCR amplification is used for mtDNA
enrichment.

**Discussion**

Mitochondrial DNA from the un-adapted version of this method did not undergo
quantification or short-read sequencing when it was initially reported. Here, we have
described a modified version of the protocol and have quantified the increased level
of mitochondrial DNA post-enrichment in 7 different mouse tissues. This method will
enable researchers to identify changes in low frequency heteroplasmy without
introducing PCR biases or NUMT contamination that are falsely identified as
heteroplasmy when long-range PCR is used.

**Abbreviations**

NUMT – Nuclear mitochondrial sequences

mtDNA – Mitochondrial DNA

POLG – Polymerase gamma

UMI – unique molecular identifier

**Introduction**

[revised manuscript text omitted]

observed across all samples using both enrichment methodologies. Average
alternative allele frequency displayed a different pattern of results compared to the
previous two heteroplasmy metrics. $Polg^{D257A}$ tissues had lower mean alternative allele
frequencies than $Polg^{wt}$. Significantly higher mean alternative allele frequencies were
observed in IrPCR tissues from both male and female $Polg^{D257A}$ mice. Conversely,
$Polg^{wt}$ samples enriched using the mtDNA prep methodology had higher mean
alternative allele frequency than those enriched via IrPCR. Tissues are highlighted by

colour, each line represents the same tissue that was enriched using both
methodologies. Statistical comparisons between IrPCR and mtDNA prep were
performed using a Wilcoxon signed-rank test. There were 24 IrPCR samples and 24
mtDNA prep samples (*Polg*^{wt} n=24, *Polg*^{D257A} n=24). Two samples for each tissue, sex
and genotype were analysed.

**c**, Heteroplasmy levels of C57BL6 wild-type mice using both IrPCR and mtDNA prep
enrichment methods. Mitochondrial DNA from C57BL6 tissues that were enriched
using IrPCR had substantially higher levels of heteroplasmy across the three assessed
metrics; top, number of heteroplasmic sites, middle, cumulative heteroplasmic burden
and bottom, average heteroplasmy. Unlike the *Polg* mutator mouse comparisons,
these enrichments were not performed on the same tissues however the mice were
the same sex and age at sacrifice. Statistical comparisons between IrPCR and mtDNA
prep were performed using a paired Student's T-test. There were six samples in the
IrPCR group and twelve samples in the mtDNA prep group.

**Supplementary figure 1. The distribution of nuclear contamination across the genome**
**shown as boxplots for each tissue.**

**Supplementary figure 2. Sequence alignment results between NUMT on chromosome**
**1 and its homologous region on chromosome 1 which shows 99.94% sequence**
**identity.**

**Supplementary figure 3. Integrated genome viewer (IGV) image of high-coverage**
**region on chromosome 2. This image shows the region that reads have aligned to on**
**chromosome 2 due to its sequence similarity with other regions in the genome.**

**Supplementary figure 4. BLAST results table that shows the high level of sequence**
**identity between the repetitive region on chromosome 2 and similar regions on other**
**chromosomes in the mouse genome.**

**Supplementary figure 5. Integrated genome viewer (IGV) image of high-coverage**
**region on chromosome 9. This image shows the region that reads have aligned to on**
**chromosome 9 due to its sequence similarity with other regions in the genome.**

Supplementary figure 6. BLAST results table that shows the high level of sequence
identity between the repetitive region on chromosome 9 and similar regions on other
chromosomes in the mouse genome.

Supplementary figure 7. The difference in number of heteroplasmic sites found
between IrPCR and mtDNA preparations in *Polg*^{D257A/D257A} and *Polg*^{wt/wt} tissues shown
as boxplots.

Supplementary figure 8. The number of heteroplasmic sites identified in each tissue
of *Polg*^{D257A/D257A} and *Polg*^{wt/wt} mice (n=4). Wilcoxon rank-sum test.

Supplementary figure 9. The tissue-specific pattern of mutation across *Polg*^{D257A/D257A}
and *Polg*^{wt/wt} mice. a, Kidney, liver, colon, heart and lung had a similar number of
heteroplasmic sites in *Polg*^{D257A/D257A} mice with higher levels observed in spleen and
lower levels found in brain. b, Colon and spleen had higher levels of average
heteroplasmy than the other five tissues in *Polg*^{D257A/D257A} mice. c, Cumulative
heteroplasmic burden was higher in *Polg*^{D257A/D257A} colon and spleen tissues and lower
in brain than in heart, kidney, liver and lung. Wilcoxon rank-sum test with kidney acting
as comparator for each statistical test.

Supplementary figure 10. Characterisation of variants that were found across all
samples using both IrPCR and Mito-SiPE. a, Long-range PCR amplification displayed
similar results to Mito-SiPE in terms of proportion of mutations which occurred at each
nucleotide in the mitochondrial genome. *Polg*^{D257A/D257A} mice had a much larger
number of mutations compared to *Polg*^{wt/wt} and C57Bl6 wild-type mice. There was a
significant difference in the mutations found in C57Bl6 between IrPCR and Mito-SiPE.
b, Mutations that were present at a frequency $\geq 10\%$ heteroplasmy showed similar
mutational profiles between IrPCR and Mito-SiPE. Although the Ts/Tv ratios appeared
to be slightly different, this was not statistically significant (Chi-squared, $p=0.43$).

Supplementary figure 11. a, The difference in number of heteroplasmic sites identified
using both methods plotted by location. The x axis represents the location in the
mitochondrial genome (grouped as 100bp regions) and the y axis represents the

difference in the number of heteroplasmic sites found in all samples using lrPCR and
Mito-SiPE. Long range PCR amplification has more variants identified across the
whole genome with a marked increase at locations where sequencing depth is
reduced. The red line represents the average difference between the methods across
the whole mitochondrial genome. b, Annotation of the variants using snpEff. Variants
were annotated and characterised by their predicted effect. There was no obvious
pattern between the difference of high, medium and low impact variants; however,
more variants were identified using lrPCR in mt-Nd1 and mt-Nd4 genes. These genes
are located in a region that has lower sequencing depth using lrPCR than Mito-SiPE.

Table 1. The number of mitochondrial, nuclear and unmapped reads per tissue.

Number of reads (millions)

Tissue	Mitochondrial (sd)	Nuclear (sd)	Unmapped (sd)
Brain	20.027 (3.285)	2.575 (2.729)	0.073 (0.178)
Heart	19.636 (2.166)	2.188 (1.481)	0.042 (0.099)
Kidney	17.428 (4.224)	4.783 (4.344)	0.012 (0.021)
Liver	18.931 (4.071)	3.291 (3.774)	0.081 (0.171)
Lungs	12.206 (2.938)	9.860 (3.137)	0.057 (0.134)
Muscle	13.146 (3.467)	9.093 (3.589)	0.156 (0.252)
Spleen	13.813 (3.827)	9.301 (4.368)	0.037 (0.072)
All Tissues	16.641 (4.586)	5.700 (4.660)	0.057 (0.139)

Table 2. The percentage of reads aligned to the mitochondrial genome, nuclear
genome and unmappable reads.

Percentage of total reads

Tissue	Mitochondrial (sd)	Nuclear (sd)	Unmapped (sd)
Brain	88.64 (11.09)	11.02 (11.20)	0.34 (0.85)
Heart	89.93 (6.19)	9.87 (6.23)	0.21 (0.50)
Kidney	78.68 (18.60)	21.26 (18.63)	0.06 (0.10)
Liver	85.14 (15.36)	14.52 (15.43)	0.33 (0.63)
Lungs	55.30 (13.21)	44.44 (13.30)	0.26 (0.60)
Muscle	58.90 (15.02)	40.39 (14.98)	0.71 (1.17)
Spleen	60.15 (16.53)	39.69 (16.54)	0.16 (0.32)
All Tissues	74.64 (19.76)	25.10 (19.78)	0.26 (0.63)

Table 3. The average total number of reads, mapped reads and sequencing depth for
 each methodology and genotype.

Assay type	Genotype	Total reads		Mapped reads		Coverage	
		Average (10 ⁷)	SD	Average (10 ⁷)	SD	Average (10 ⁵)	SD
IrPCR amplification	Polg ^{D257A}	7.22	0.561	7.19 (99.5%)	0.50	1.39	0.417
mtDNA prep	Polg ^{D257A}	7.15	0.848	3.98 (55.7%)	1.86	1.22	0.245
IrPCR amplification	Polg ^{wt}	7.19	0.687	7.08 (98.5%)	0.693	1.36	0.411
mtDNA prep	Polg ^{wt}	7.02	0.687	4.53 (64.5%)	1.79	1.25	0.126

Methods

Breeding and tissue harvesting

All animal protocols were reviewed and approved by the National Human Genome
 Research Institute (NHGRI) Animal Care and Use Committee prior to animal
 experiments. Mice were housed in shoe box cages and fed ProLab RMH 1800 diet
 (PMI Nutrition International) containing 50 µg vitamin B12/kg of diet and 3.3 mg folic
 acid/kg of diet. Breeding mice were fed Picolab Mouse Diet 20, containing 51 µg
 vitamin B12/kg diet and 2.9 mg folic acid/kg of diet. Heterozygous *Polg*^{wt/D257A} males
 and *Polg*^{wt/D257A} females were mated. Homozygous mutant and wild-type progeny
 were aged to 6 months at which point they were sacrificed. There were 4 *Polg*^{wt/wt} mice
 (2 male, 2 female) and 4 *Polg*^{D257A/D257A} mice (2 male, 2 female) used in the *Polg*
 experiments. Brain, heart, lung, liver, spleen, kidney and muscle tissue was isolated
 and mitochondrial DNA enrichment was performed on all tissues

Tissue homogenisation

Harvested tissue was placed in a homogenization tube with 10X volume per gram of
 fresh homogenization buffer i.e. 5 ml buffer for 500 mg tissue. Tissues were

homogenised until no discernible whole tissue was present. The homogenate was
then transferred to 1.5 ml microcentrifuge tubes and spun at 1000 g for 1 minute at
4°C. The supernatant was transferred to a new microcentrifuge tube and spun at 12000
618 g for 10 minutes at 4°C to pellet mitochondria. The mitochondrial pellet was
619 resuspended with 100 µl of resuspension buffer for storage or for immediate DNA
extraction.

**Mitochondrial DNA isolation**

The mitochondria resuspension was added to 200 µl alkaline lysis buffer, vortexed,
and placed on ice for 5 minutes. Potassium Acetate Buffer (150 µl) was then added
and the mixture was vortexed slowly and placed on ice for 5 minutes. The mixture was
centrifuged at 12,000 g for 5 minutes at 4°C to pellet proteins and the supernatant was
decanted to a new tube. RNase (1 µg) was added to the mixture and left at room
temperature for 15 minutes. Phenol-chloroform (500 µl) was added to each tube,
inverted and placed on a shaker/rotator for 20 minutes. Afterwards, centrifugation at
12000 g for 2 minutes at room temperature was carried out. The aqueous (top) layer
was decanted to a new tube (approx. 450 µl from this phase was retrieved) and 40 µl
sodium acetate, 1 µl glycogen (20 mg/ml) and 1200 µl 100% EtOH were added. The
mixture was inverted and mixed well then left on dry ice for 60 minutes. The mixture
underwent centrifugation at 12,000 g and the supernatant was removed. The pellet
was finally washed twice using 70% ethanol, air-dried, and resuspended in a low-TE
buffer for sequencing or regular TE buffer for (q)PCR.

**Library preparation and next generation DNA sequencing**

Libraries were generated from approximately 50 ng genomic DNA using the Accel-
NGS 2S Plus DNA Library Kit (Swift Biosciences) using 5 cycles of PCR to minimize
PCR bias. The DNA samples were sheared by sonication (Covaris Inc., Woburn, MA)
to a mean of 300 bp. Libraries were tagged with unique dual index DNA barcodes to
allow pooling of libraries and minimize the impact of barcode hopping. Libraries were
pooled for sequencing on the NovaSeq 6000 (Illumina) to obtain at least 7.6 million
151-base read pairs per individual library. Sequencing data was processed using RTA
version 3.4.4.

**Data processing and alignment**

Fastq files were aligned to the mouse reference genome, GRCm38, using bwa mem
using the default parameters⁴⁹. Picard tools were used to add read groups, and to
mark and remove duplicates⁵⁰. Samtools was used to calculate the coverage across
the nuclear and mitochondrial genome for each sample. Finally, R (v3.5.0) and ggplot2
(v3.3.0) were used for statistical analysis and subsequent visualisation of graphs^{51,52}.
**Library complexity and fragment sizes were calculated using Picard tools v 1.4.2 on**
**15 randomly-selected samples (Supplementary table 2).**

**Variant calling and mutation analysis**

**Variant calling was performed using bcftools v1.9 with 'bcftools mpileup -f -Q 30 -skip-**
**indels reference_fasta bam_file | bcftools call -mv' to identify single nucleotide variants**
**only. Filtering was performed by removing any SNVs that had a QUAL score lower**
**than 20. The code used for alignment and variant calling is available on github**
**(<https://github.com/walshd59/mtDNAhetScripts.git>).** Of 66,738 variants identified
across all samples in our study at an alternative allele frequency $\geq 0.2\%$, only 137 of
these had a $\ln(\text{Strand Odds Ratio})$ value ≥ 3 . **Analysis of the mutation spectrum and**
**further characterisation/annotation of heteroplasmic variants was performed using**
**SnEff (v 5.1)⁵³.**

**Quantification of mtDNA copy number**

Mitochondrial DNA copy number was assessed via qPCR targeting both mt-16S and
nuclear-encoded hexokinase (HK), similarly as previously described⁵⁴. Briefly, 2.5 μl
LightCycler® 480 SYBR Green I Master (Roche, Molecular Systems, Inc, Germany),
2 μl of DNA (20 ng/ μl) and 0.5 μl primer mix were added in triplicate to a 384-well plate
and the reactions were carried out by the QuanStudio 6 Flex (Applied Biosystems,
Foster City, CA, USA). The conditions were as follows: 95°C for 5 min, 45 cycles of
95°C for 10 s, 60°C for 10 s and 72°C for 20 s. A melting curve was performed using
95°C for 5 s, 66°C for 1 min and gradual increase to 97°C. Mitochondrial DNA copy
number was assessed using the following formula: $2 \times 2^{\Delta\text{Ct}}$ where $\Delta\text{Ct} = \text{Ct}(\text{mtDNA}$
$\text{gene}) - \text{Ct}(\text{nDNA gene})$.

**Long-range PCR enrichment of mtDNA**

This technique was used to amplify human and mouse mitochondrial DNA in two
fragments from a whole DNA extract. DNA was quantified *via* Nanodrop (Methods
2.2.7) unless otherwise stated. Each PCR reaction consisted of Q5 High-fidelity
Polymerase (0.02 U/ μ l), 5X Q5 reaction buffer (1X), 10mM dNTPs (300 μ M), 5 μ M
Forward and Reverse primers (0.25 μ M, Table 2.3; human, Table 2.4; mouse).
Template DNA (100 ng) was added to each reaction except for the no-template control
(NTC) but an equivalent volume of molecular biology grade water was added instead.
The temperature cycles were as follows: 1 x 30 s denature 98°C, 25 x 10 s denature
98°C, 30 s annealing 66°C, 4 minutes 30 s elongation 72°C, 1 x 10 minutes elongation
72°C on a thermocycler. Both fragments were quantified using qubit and mixed in
equimolar ratios.

**Statistics and reproducibility**

All statistical analyses included in this paper were carried out in R (version 4.1.1) and
the software package rstatix (version 0.7.0). Sample sizes are described within each
experimental figure.

**Solutions**

Homogenization Buffer:	0.25 M Sucrose, 10 mM EDTA, 30 mM Tris-HCl, 700 pH=7.5.
Resuspension Buffer:	10 mM Tris, 0.15 M NaCl, 10 mM EDTA, pH=8.0.
Alkaline Lysis Buffer:	0.18 N NaOH, 1% SDS (Prepare fresh).
Potassium Acetate Buffer:	3M potassium, 5M acetate.
Low TE buffer:	10 mM Tris-HCl, 0.1 mM EDTA, pH=8.0.

Ethics approval and consent to participate

Not applicable

Consent for publication

Not applicable

Availability of data and materials

All sequencing data and associated metadata is available on SRA (PRJNA881035).

Competing interests

The authors declare that they have no competing interests

Funding

This work was funded by Wellcome Trust, National Human Genome Research Institute and NIH Intramural Sequencing Centre.

Author Contributions

DW, ME and DB performed the sample collections and enrichment. DW and DH performed bioinformatics and subsequent analysis. DW, DB, FP, DH, APMcD and LB prepared 
[revised manuscript text omitted]

REVIEWERS' COMMENTS:

Reviewer #1 (Remarks to the Author):

I appreciate the efforts that the authors have made to revise the method and encourage the acceptance of the manuscript in its current form. Congratulations on the nice contribution.

Reviewer #2 (Remarks to the Author):

In the revised manuscript, Walsh et al. discussed the advantage of their method over Gould et al. They have substantiated their claim regarding the lost coverage in the middle of the mtDNA. Additionally, they have clarified the parts of the manuscripts that were not clear in the first submission.

I firmly believe this manuscript will be of great interest to the readers, and I recommend accepting this work for publication.